# A comparative study on the intensity of loneliness among Kenyan youth in school and home environments

Victoria Mutiso[1], David Ndetei[2,3] , Christine Musyimi[1], Pascalyne Nyamai[1], Denis Kioko[1], Diana Thakya[1], Kevin Onuonga[2], Susan Malinda[2], Yvonne Kiogora[1] , Diana Achola[1], Samuel Walusaka[1], Veronica Onyango[1], Eric Jeremiah[1], Andre Sourander[4] and Daniel Mamah[5]

[1]Africa Institute of Mental and Brain Health, Kenya; [2]Psychiatry, Africa Institute of Mental and Brain Health, Kenya; [3]Psychiatry, University of Nairobi, Kenya; [4]University of Turku, Finland and [5]Washington University in St Louis, USA

## Research Article

**Keywords:**
loneliness; UCLA loneliness scale; school environment; home environment; demographic factors

**Corresponding author:**
David Ndetei;
Email: dmndetei@amhf.or.ke

## Abstract

Loneliness is a public health concern influenced by environmental contexts. Among youth, it manifests differently at school and home, yet research in low-resource settings is limited. This study examined patterns of loneliness and how economic and sociodemographic factors correlate with it in school and home environments among Kenyan youths in the Nairobi Metropolitan Area. A cross-sectional study was conducted with 1,972 youths aged 14–25 years using a self-administered questionnaires. Analyses included paired t-tests, ANOVA and generalized estimating equations (GEEs). Among participants with complete paired data (n = 1,166), loneliness was significantly higher at school (M =23.15) than at home (M = 21.53). Females reported higher loneliness than males (school: p=.011; home: p<.001). Education level and marital status were significantly related to loneliness at home (p<.001 and p=.022) but not at school. Loneliness at home was higher among the poorest households compared to middle-class households (mean difference =2.556, p=.048). GEE models confirmed these patterns and indicated that employment status influenced differences in loneliness between home and school settings. School settings were linked with higher loneliness, while home loneliness varied by socioeconomic and demographic factors, underscoring the need for targeted interventions addressing environmental and social determinants of youth loneliness.

## Impact statement

This study offers a critical examination of the contextual, economic and sociodemographic dimensions of loneliness among Kenyan youths, showing how these experiences differ between school and home settings. By identifying both protective and risk factors within these environments, the findings underscore the need for interventions that are sensitive to transitional stressors, family dynamics and cultural realities. The results provide valuable insights for policymakers, educators, mental health practitioners and community stakeholders concerned with youth well-being. Beyond contributing to academic literature, this study highlights that loneliness is a multidimensional social issue that demands coordinated responses across education systems, family structures and public policy frameworks.

## Introduction

Loneliness has increasingly been described as a pressing concern globally because of its wide-ranging consequences for psychological well-being and physical health. It has been associated with depression, anxiety, weakened immunity, cardiovascular disease and reduced life expectancy (Garcia et al., 2025). Global studies frequently highlight that loneliness affects people across the life span, although its intensity and frequency vary depending on age, gender, socioeconomic status and cultural background (Barreto et al., 2021). While many surveys suggest rising levels of loneliness, particularly among adolescents and older populations, some longitudinal analyses do not suggest uniform findings but suggest context-dependent findings, shaped by both structural and cultural factors (Mund et al., 2020).

Education is often cited as a determinant of loneliness, but the evidence remains inconsistent. Higher levels of education are usually linked with larger networks, stronger social resources and greater coping capacities that help reduce feelings of isolation (Balki et al., 2023). However, on the other hand, numerous studies show that students in higher learning institutions report heightened loneliness, often explained by the stresses of relocation, intense academic expectations and

identity transitions (Walsham et al., 2023). Similarly, among older adults, the role of education is less clear, with some research showing little or no direct impact on loneliness once other socio-economic variables are controlled for (Sánchez-Moreno et al., 2024). Employment patterns reveal a similar inconsistent findings. Some studies have suggested that being employed is consistently associated with higher loneliness due to financial insecurity, lack of routine and reduced social participation (Bryan, 2024; Smith and Eng, 2024). Further, employment itself does not always guarantee protection, as jobs marked by instability, excessive pressure or limited social interaction may actually intensify loneliness (DIMITRIU, 2022). Flexible or remote arrangements can likewise be supportive for some and isolating for others, depending on the availability of supportive networks (Rogers, 2022). Marital and relational status emerges as another central dimension. Married or cohabiting individuals generally support lower loneliness compared to those who are never married, divorced or widowed (Kislev, 2022). Nonetheless, poor-quality partnership can create deep loneliness even within marriage, while single individuals with robust friendship and community ties may thrive socially (Garcia et al., 2025). Previous studies indicate that lower household wealth is associated with higher levels of loneliness, suggesting that economic disadvantage may limit opportunities for social engagement and supportive interactions (Algren et al., 2020). Some studies underscore the relationship between socioeconomic conditions and loneliness with basic utilities such as electricity, piped water and gas consistently linked with lower loneliness scores (Reed and Bohr, 2021; Lu et al., 2025). However, not all indicators of socioeconomic status directly influence loneliness. Structural measures, such as flooring type, sanitation facilities or possession of multiple household assets, often show weak or no correlation with feelings of isolation (Wee et al., 2019).

In the African context, research on loneliness is expanding and reveals dynamics shaped by rapid social change. Traditionally, extended family systems and community-based living were assumed to protect individuals against isolation (Głąb and Kocejko, 2024). However, urbanization, labor migration and shifting cultural norms have weakened some of these structures, leading to new vulnerabilities. Studies across sub-Saharan Africa show that loneliness is closely tied to socioeconomic hardship, unemployment and the breakdown of communal bonds (Paugam, 2016; Adedeji et al., 2023). Adolescents and young adults have been identified as especially at risk, with loneliness often intertwined with stigma, mental health challenges and limited opportunities for social mobility. Additionally, gender continues to play a significant role, with females often reporting higher emotional distress linked to social exclusion (Smit, 2021). Other evidence from certain rural or community-based settings indicate that collective caregiving practices and strong kinship ties can still act as buffers, showing that the African experiences of loneliness is neither uniform nor entirely negative (Morobane, 2020). These contrasting findings highlight the need to interpret loneliness in Africa through the lens of changing cultural and structural realities. A study in South Africa demonstrated a relationship between socioeconomic status and loneliness, finding that individuals who perceived their household economic status as below average experienced higher levels of loneliness, whereas those who rated their economic status as average reported significantly fewer episodes of loneliness (Posel, 2025). Access to communication technologies, such as televisions, radios and mobile phones, shows mixed outcomes, reflecting varying patterns of usage, affordability and cultural engagement (Schroeder, 2010).

Kenyan research echoes some of these regional patterns while also highlighting unique national dimensions. Adolescents and young adults are frequently reported to experience loneliness, with studies linking it directly to depression, hopelessness, suicidality and post-traumatic stress symptoms (Ndetei et al., 2025). Unemployment, particularly among youth, emerges as a consistent predictor, reinforcing the connection between economic vulnerability and social isolation (Redmond et al., 2024). Shifts in family structures, including reduced parental involvement and weaker extended family support, have also been associated with increased feelings of disconnection among young people (Branje and Morris, 2021). Yet, in rural parts of the country and within strong community networks, social belonging remains a protective factor (Ellis et al., 2015). Investigations in Kenya emphasize that loneliness is not simply an individual condition but one tied closely to structural issues such as poverty, migration and educational transitions (Koogler, 2024). Despite growing research, most studies focus on overall prevalence and broad correlates, with limited evidence on how loneliness varies across everyday social environments.

Most of the adolescents spend their time in school and home settings, each presenting distinct social demands. Schools may expose youth to academic pressure, peer comparison, bullying and social exclusion, while homes vary in emotional support, supervision and stability. Yet no Kenyan study has directly compared loneliness across these two contexts within the same individuals. This study addresses this gap by quantifying and comparing loneliness at school and at home and identifying demographic and socioeconomic correlates that may operate differently across these settings.

## General objective

To examine the patterns of loneliness and how economic and sociodemographic factors correlate with it in school and home environments among Kenyan youths living in the Nairobi Metropolitan Area.

## Specific objective

1. To determine the patterns of loneliness in school and home environments in Kenyan youths living in the Nairobi Metropolitan Area.
2. To assess the influence of economic and sociodemographic factors on loneliness in the school and home environment in Kenyan youths living in the Nairobi Metropolitan Area.

## Methods

### Study design and setting

This population-based cross-sectional study was conducted from September 21, 2022 to December 15, 2022. It was done in an urban and peri-urban location in the Nairobi Metropolitan Area. To ensure diversity and representation, adolescents from economically disadvantaged neighborhoods within the Nairobi Metropolitan Area were purposively included, allowing for a comparative perspective on how contrasting social and environmental conditions contribute to experiences of loneliness.

### Study participants

The study involved youths aged 14–25 years residing in selected neighborhoods within Nairobi's Metropolitan Area. Participants were recruited from both colleges and the general community. The

general community included youth who were still in school and those who were out of school. In college settings, administrative permission was obtained, and students were approached through organized group sensitization sessions. In community settings, recruitment was facilitated through local Chiefs and community opinion leaders, who used a standardized script to inform eligible youths about the study purpose, procedures and voluntary nature of participation. Participants reported their experiences of loneliness in two distinct contexts: at school (for those attending school or college) and at home (for all participants, including those out of school). Inclusion required the ability to understand the study and provide informed consent. Exclusion criteria included participants unable to comprehend the questionnaire due to intoxication at the time of the interview, inability to read the questionnaire language, being outside the specified age range and unwillingness to participate. This selection process ensured ethically sound participation and a representative sample for the study's focus on loneliness.

### Procedures

#### Research assistant selection and training

Twelve research assistants (RAs) were recruited through a competitive process designed to assess prior experiences and interpersonal skills. Subsequently, successful candidates underwent a 2-day comprehensive in-person training, involving data collection techniques, protocols and simulated data collection role plays.

#### Community engagement and recruitment

Approvals were secured through the County Commissioners from the two sites. We then worked with Chiefs to mobilize and raise awareness among youth through community opinion leaders about the study. Community opinion leaders used a standardized information script to sensitize potential participants, explain the goals and value of the study and encourage broad participation, including marginalized groups. This participatory approach leveraged community trust, reduced potential barriers and minimized bias.

We provided the Chiefs and the community opinion leaders with a detailed script that explained the study's purpose, importance and goals. Using this script, they actively engaged youth, organized participation and encouraged involvement by leveraging existing trust and social connections. The script also included an invitation with details on the date, time and venue. This approach ensured representation of youths across different socioeconomic backgrounds, including high-risk populations and marginalized groups, improving the relevance of our findings and minimizing selection bias.

Sample Achieved: No financial or material incentives were provided. A total of 1,972 participants participated in the study after providing written consent. None of the participants declined to participate (consistent with our previous studies, where the response rate has been nearly 100% (Ndetei et al., 2010).

#### Data collection

A self-administered questionnaire consisting of the University of California, Los Angeles (UCLA) loneliness scale, sociodemographic and wealth index sections was administered in a group setting within social halls at the study sites to facilitate efficient data collection. The process began by randomly assigning numbers (1–12) to RAs, with each number forming a distinct group. Participants were then allocated to groups through a restricted randomization process using printed vouchers marked with numbers 1–12, ensuring evenly distinct group sizes. Each group consisted of a maximum of 25 participants and was supervised by a trained RA. Participants completed the assessment independently, reading and responding to the questions on their own. The group setting was because of logistical convenience, whereby the RA was available to oversee and to ensure that there was no communication between the subjects. Confidentiality was strictly maintained, and participants were not allowed to discuss or disclose their answers to others. Similar to a national examination setting, anonymity was preserved to reduce potential stigma, with the RA ensuring that no discussions took place during the session. This was a fully participant-administered questionnaire. The role of an RA was limited to explaining the nature and purpose of the study and overseeing the session. Written informed consent was obtained before participation, and RAs cross-checked the participants' ages to confirm eligibility based on the study's inclusion criteria.

### Tools

#### Sociodemographic profile

A self-reported sociodemographic questionnaire was used to gather information on age, gender, marital status, religion, level of education and employment status.

#### Wealth index

Wealth Index was used to assess family socioeconomic status, based on household assets, water sources, floor type, toilet facilities and primary cooking fuel, following the World Bank model for low- and middle-income countries. It was constructed using principal component analysis from the Demographic Health Survey framework and classified on a 5-point Likert scale from high to low (McLorg et al., 2021).

#### UCLA loneliness scale

The UCLA Loneliness Scale developed by Russell et al. (1978) is 20-item self-report questionnaire used to measure individuals' subjective experiences of loneliness and social isolation (Russell et al., 1978). The scale was administered twice to each participant to assess contextual differences. Participants were instructed to respond based on their experiences "at school" (or college), and subsequently based on their experiences "at home." This approach allowed for within-subject comparison of loneliness across settings while maintaining measurement consistency.

Participants rated each item as either 1 ("I often feel this way"), 2 ("I sometimes feel this way"), 3 ("I rarely feel this way") and 4 ("I never feel this way"). For analysis, responses were recorded in line with the scale's scoring system, where "Often" = 3, "Sometimes" = 2, "Rarely" = 1 and "Never" = 0. The total score ranges from 0 to 60, with higher scores indicating greater levels of loneliness. Although the scale does not have fixed clinical cut-off points, scores are typically interpreted using group means, quartiles or percentiles. The UCLA Loneliness Scale has shown strong psychometric properties, including high internal consistency (α > 0.89) and strong test–retest reliability, particularly among adolescent and young adult populations (Russell et al., 1978; Solano, 1980). It is widely used in psychological and sociological research for evaluating emotional and social well-being (Park and Berkowitz, 2024).

### Statistical analysis

The data were analyzed using IBM SPSS Statistics version 25. Descriptive statistics, including frequencies and percentages, were first computed to provide an overview of participants' responses on

loneliness across school and home contexts. Chi-square, ANOVA and t-tests were used to test for significant differences in loneliness in school and home, and across the selected economic and socio-demographic factors. To enhance the robustness of the t-test and ANOVA results and account for potential deviations from normality, bootstrapping with 5,000 resamples was applied to generate bias-corrected confidence intervals. Where ANOVA results were significant, post hoc tests (Bonferroni) were conducted to identify specific group differences. These statistical procedures were chosen because they are appropriate for categorical data and allow for the identification of patterns, group differences and associations within the sample. All tests were conducted at a 95% confidence level, with the threshold for statistical significance set at p < 0.05. A generalized estimating equation (GEE) model was used to examine predictors of loneliness while accounting for the repeated measures across home and school settings. The model included loneliness score as the dependent variable, loneliness setting (home vs. school) as a within-subject factor, and age, gender, education, marital status, religion and employment as predictors. Interaction terms between setting and each predictor were included to test whether associations differed by context, with an exchangeable working correlation structure, identity link and normal distribution. In addition, the GEE framework provides robust standard errors that reduce bias arising from potential clustering of participants within schools or recruitment sites. This approach provides population-averaged estimates that are robust to mild violations of normality and within-subject dependence, thereby strengthening the validity of inferences drawn from the paired analyses.

### Ethics

This study was reviewed and approved by the Nairobi Hospital Ethics Research Committee (TNH-ERC/DMSR/ERP/022/22) and licensed by the National Commission for Science, Technology and Innovation (NACOSTI) license number NACOSTI/P/22/18097. The study was in accordance with the Helsinki Declaration of 1975, as revised in 2008. Administrative permissions were sought from the county-level offices, as well as institutional approval was obtained from the participating colleges prior to data collection. Written informed consent for youths aged 18 years and above was obtained before data collection commenced. For minors (under 18 years), consent was obtained from their parents or legal guardians. Minors also provided written assent, and when they declined, they were not enrolled, even if parental consent had been granted. The participants were also informed of the risk that some of the questions may evoke memories or trigger difficult emotions. However, they were allowed to pause, skip or withdraw from the study at any time without penalty. School and community participants who experienced discomfort during or after participation were referred to their school counselor and RAs, respectively, who provided initial psychosocial support and guidance. They were further referred to the nearest health facility for evaluation and ongoing care. Data were anonymized at collection, stored on password-protected and encrypted devices, accessible only to authorized researchers.

### Results

### UCLA loneliness scale

Across all 20 items, there was a statistically significant difference between home and school settings ($\chi^2$ range = 194.68–568.61, all

p < .001), showing that the experience of loneliness was higher at school compared to the home setting (Table 1).

There were variations in percentage loneliness across the UCLA items. Most participants at school (57.8%) as compared to 51.1% at home reported feeling unhappy doing many things alone. Similarly, 44.7% of students indicated they could not tolerate being so alone at school, compared to 40.6% at home. This also reflected in the quality of relationships that students felt about their relationships. 53.9% of students reported that no one really knows them well at school, compared to 44.1% at home. Similarly, 47.7% believed their social relationships at school were superficial as well as 44.9% at home. When asked whether people were around them but not truly with them, 47.1% of students agreed at school versus 40.5% at home. Feelings of being shut out and excluded were also more pronounced in the school setting, with 38.6% of students experiencing this at school compared to 32.1% at home.

### Comparison of loneliness scores between school and home

From Table 2, among participants who completed the paired sample t-test (n = 1,166), the results showed a higher mean loneliness score at school (23.15) than at home (21.53). The difference was statistically significant, with t(1,165) = 5.3, p < .001, Cohen's d = 0.16. The 95% confidence interval for the difference between means ranged from 1.02 to 2.22, indicating that the true mean difference lies within this range. The correlation coefficient of r = 0.651 indicated a strong statistically significant relationship between loneliness scores in the two settings.

### Social demographics and loneliness

Gender was significantly associated with loneliness, with females reporting higher loneliness scores than males at both school (female: 23.93, male: 22.08, p = .011) and home (female: 23.21, male: 21.13, p < .001), indicating a consistent pattern across settings. In contrast, religion and employment status did not show statistically significant differences in loneliness scores in either setting, suggesting these factors had no notable impact on reported loneliness (see Table 3).

Education level was not significantly associated with loneliness at school (p = .145) but showed a significant relationship at home (p < .001). In a further test on loneliness at home versus education (post hoc comparison), participants with primary or secondary education reported higher loneliness scores than those with tertiary education. Similarly, marital status was not significantly associated with loneliness at school (p = .181) but was significant at home (p = .022). Higher loneliness scores were observed among individuals who were separated, divorced or married/cohabiting compared to those who were never married (see Table 4). When the sample was stratified by age among adolescents aged 14–18, marital status was not significantly associated with loneliness (F(6,341) = 1.020, p = 0.412). In contrast, among young adults aged 19–25, marital status was significantly associated with loneliness (F(6,1,454) = 2.898, p = 0.008) (see Supplementary Table 1).

Further analysis on the interaction between gender and social demographics on loneliness shows that, at home, gender significantly interacted with marital status (p = 0.038) and employment (p = 0.033), indicating that the effect of these variables on loneliness differed by gender. At school, there were no significant interactions (see Supplementary Table 2).

**Table 1.** Comparison of loneliness between school and home settings

| Variable | Setting | Never n (%) | Rarely n (%) | Sometimes n (%) | Often n (%) | Chi-square value, degrees of freedom (df) and p-value |
|---|---|---|---|---|---|---|
| 1. I am unhappy doing so many things alone | School | 231 (21.0%) | 233 (21.2%) | 322 (29.3%) | 314 (28.5%) | $\chi^2$ (9) = 194.68, p < .001 |
| | Home | 321 (29.2%) | 216 (19.6%) | 271 (24.6%) | 292 (26.5%) | |
| 2. I have nobody to talk to | School | 477 (43.4%) | 228 (20.7%) | 235 (21.4%) | 160 (14.5%) | $\chi^2$ (9) = 313.942, p < .001 |
| | Home | 511 (46.5%) | 210 (19.1%) | 202 (18.4%) | 177 (16.1%) | |
| 3. I cannot tolerate being so alone | School | 388 (35.4%) | 219 (20.0%) | 242 (22.1%) | 248 (22.6%) | $\chi^2$ (9) = 484.117, p < .001 |
| | Home | 412 (37.6%) | 236 (21.5%) | 241 (22.0%) | 208 (19.0%) | |
| 4. I lack companionship | School | 483 (43.8%) | 221 (20.0%) | 206 (18.7%) | 194 (17.6%) | $\chi^2$ (9) = 389.715, p < .001 |
| | Home | 517 (46.8%) | 207 (18.8%) | 187 (16.9%) | 193 (17.5%) | |
| 5. I feel as if nobody really understands me | School | 374 (34.4%) | 199 (18.3%) | 287 (26.4%) | 227 (20.9%) | $\chi^2$ (9) = 491.051, p < .001 |
| | Home | 390 (35.9%) | 198 (18.2%) | 268 (24.7%) | 231 (21.3%) | |
| 6. I find myself waiting for people to call or write | School | 387 (35.0%) | 187 (16.9%) | 276 (25.0%) | 255 (23.1%) | $\chi^2$ (9) = 436.302, p < .001 |
| | Home | 424 (38.4%) | 210 (19.0%) | 250 (22.6%) | 221 (20.0%) | |
| 7. There is no one I can turn to | School | 525 (47.7%) | 188 (17.1%) | 198 (18.0%) | 189 (17.2%) | $\chi^2$ (9) = 358.562, p < .001 |
| | Home | 597 (54.3%) | 177 (16.1%) | 169 (15.4%) | 157 (14.3%) | |
| 8. I am no longer close to anyone | School | 548 (50.4%) | 189 (17.4%) | 181 (16.7%) | 169 (15.5%) | $\chi^2$ (9) = 398.128, p < .001 |
| | Home | 598 (55.0%) | 177 (16.3%) | 159 (14.6%) | 153 (14.1%) | |
| 9. My interests and ideas are not shared by those around me | School | 404 (36.6%) | 221 (20.0%) | 298 (27.0%) | 182 (16.5%) | $\chi^2$ (9) = 443.583, p < .001 |
| | Home | 428 (38.7%) | 214 (19.4%) | 264 (23.9%) | 199 (18.0%) | |
| 10. I feel left out | School | 495 (45.0%) | 169 (15.3%) | 269 (24.4%) | 168 (15.3%) | $\chi^2$ (9) = 458.191, p < .001 |
| | Home | 548 (49.8%) | 185 (16.8%) | 215 (19.5%) | 153 (13.9%) | |
| 11. I feel completely alone | School | 536 (49.1%) | 180 (16.5%) | 214 (19.6%) | 161 (14.8%) | $\chi^2$ (9) = 412.883, p < .001 |
| | Home | 591 (54.2%) | 163 (14.9%) | 173 (15.9%) | 164 (15.0%) | |
| 12. I am unable to reach out and communicate with those around me | School | 470 (42.5%) | 231 (20.9%) | 241 (21.8%) | 164 (14.8%) | $\chi^2$ (9) = 419.300, p < .001 |
| | Home | 512 (46.3%) | 209 (18.9%) | 214 (19.3%) | 171 (15.5%) | |
| 13. My social relationships are superficial | School | 339 (31.0%) | 234 (21.4%) | 287 (26.3%) | 232 (21.2%) | $\chi^2$ (9) = 568.607, p < .001 |
| | Home | 381 (34.9%) | 221 (20.2%) | 272 (24.9%) | 218 (20.0%) | |
| 14. I feel starved for company | School | 478 (43.3%) | 210 (19.0%) | 237 (21.4%) | 180 (16.3%) | $\chi^2$ (9) = 453.468, p < .001 |
| | Home | 512 (46.3%) | 209 (18.9%) | 210 (19.0%) | 174 (15.7%) | |
| 15. No one really knows me well | School | 327 (29.6%) | 181 (16.4%) | 294 (26.7%) | 301 (27.3%) | $\chi^2$ (9) = 427.848, p < .001 |
| | Home | 429 (38.9%) | 187 (17.0%) | 232 (21.0%) | 255 (23.1%) | |
| 16. I feel isolated from others | School | 512 (46.2%) | 206 (18.6%) | 236 (21.3%) | 154 (13.9%) | $\chi^2$ (9) = 396.238, p < .001 |
| | Home | 564 (50.9%) | 199 (18.0%) | 188 (17.0%) | 157 (14.2%) | |
| 17. I am unhappy being so withdrawn | School | 452 (41.0%) | 225 (20.4%) | 232 (21.1%) | 193 (17.5%) | $\chi^2$ (9) = 563.024, p < .001 |
| | Home | 488 (44.3%) | 213 (19.3%) | 207 (18.8%) | 194 (17.6%) | |
| 18. It is difficult for me to make friends | School | 507 (46.0%) | 188 (17.1%) | 237 (21.5%) | 170 (15.4%) | $\chi^2$ (9) = 478.164, p < .001 |
| | Home | 545 (49.5%) | 183 (16.6%) | 206 (18.7%) | 168 (15.2%) | |
| 19. I feel shut out and excluded by others | School | 483 (44.0%) | 223 (20.3%) | 232 (21.1%) | 159 (14.5%) | $\chi^2$ (9) = 420.263, p < .001 |
| | Home | 550 (50.1%) | 194 (17.7%) | 189 (17.2%) | 164 (14.9%) | |
| 20. People are around me but not with me | School | 395 (35.7%) | 191 (17.3%) | 265 (24.0%) | 255 (23.1%) | $\chi^2$ (9) = 493.59, p < .001 |
| | Home | 464 (42.0%) | 194 (17.5%) | 238 (21.5%) | 210 (19.0%) | |

*Note:* Percentages are based on valid responses within each setting (school vs. home). Chi-square tests compare the distribution of loneliness responses across the two settings for each item. All comparisons were statistically significant at p < .001.

**Table 2.** Loneliness scores at school versus loneliness scores at home

| n | Mean (school) | Mean (home) | Std. deviation | Std. error mean | t | df | p | 95% CI lower | 95% CI upper |
|---|---|---|---|---|---|---|---|---|---|
| 1,166 | 23.15 | 21.53 | 10.44 | 0.31 | 5.3 | 1,165 | **p < .001** | 1.02 | 2.22 |

*Note:* Paired-samples t-test showed significantly higher loneliness at school than at home, t(1165) = 5.30, p < .001; scores were strongly correlated (r = .651, p < .001). Cohen's d = 0.16. Means presented are raw (unadjusted) scores. Then, 1,166 are the total participants out of 1972 with both school and home scores; remaining participants were missing one of the two measures due to incomplete responses.

**Table 3.** Demographic distribution across loneliness score at school

| Variable | Group | N = 1972 | Mean | df | t | Std. deviation | Std. error | p value | 95% CI lower | 95% CI upper |
|---|---|---|---|---|---|---|---|---|---|---|
| Gender | Female | 605 | 23.93 | 1,182.28 | 2.556 | 12.84619 | 0.52227 | 0.011 | 0.43 | 3.26 |
| | Male | 580 | 22.08 | | | 12.01389 | 0.49885 | | | |
| Age | 14–18 | 313 | 23.19 | 1,171 | 0.307 | 12.48 | 0.705 | 0.759 | −1.362 | 1.868 |
| | 19–25 | 860 | 22.94 | | | 12.47 | 0.425 | | | |
| Religion | Protestant | 528 | 23.5322 | 1,136 | 0.845 | 12.39892 | 0.53959 | 0.469 | 22.4722 | 24.5922 |
| | Catholic | 464 | 22.8621 | | | 12.67548 | 0.58844 | | 21.7057 | 24.0184 |
| | Muslim | 69 | 21.1304 | | | 13.01799 | 1.56718 | | 18.0032 | 24.2577 |
| | Others | 76 | 23.3289 | | | 11.74438 | 1.34717 | | 20.6452 | 26.0127 |
| Employment | No | 970 | 22.8237 | 1,189 | 1.217 | 12.45835 | 0.40001 | 0.302 | 22.0387 | 23.6087 |
| | No, but volunteering | 121 | 24.686 | | | 12.68072 | 1.15279 | | 22.4035 | 26.9684 |
| | Yes, working full time | 36 | 24.7778 | | | 11.47447 | 1.91241 | | 20.8954 | 28.6602 |
| | Yes, working part-time | 63 | 21.8889 | | | 12.20406 | 1.53757 | | 18.8153 | 24.9624 |
| Education | Primary | 97 | 24.0825 | 1,192 | 1.8 | 12.32517 | 1.25143 | 0.145 | 21.5984 | 26.5665 |
| | Secondary | 510 | 23.798 | | | 11.93715 | 0.52859 | | 22.7596 | 24.8365 |
| | Tertiary | 473 | 22.1501 | | | 13.15169 | 0.60472 | | 20.9618 | 23.3384 |
| | University | 113 | 22.292 | | | 11.68995 | 1.0997 | | 20.1131 | 24.4709 |
| Marital status | Never married | 805 | 23.2199 | 1,184 | 1.481 | 12.92747 | 0.45563 | 0.181 | 22.3255 | 24.1142 |
| | In a relationship, but not living together | 227 | 22.2687 | | | 11.29426 | 0.74963 | | 20.7916 | 23.7459 |
| | Married/cohabiting | 84 | 23.3452 | | | 11.72629 | 1.27944 | | 20.8005 | 25.89 |
| | Separated but not divorced | 14 | 29.6429 | | | 7.04499 | 1.88285 | | 25.5752 | 33.7105 |
| | Divorced | 7 | 24.4286 | | | 10.24463 | 3.87211 | | 14.9539 | 33.9033 |
| | Widowed | 5 | 12.4 | | | 14.63899 | 6.54675 | | −5.7767 | 30.5767 |
| | Others | 43 | 22.3721 | | | 11.83221 | 1.80439 | | 18.7307 | 26.0135 |

*Note:* Independent-samples t-tests and one-way ANOVAs examined demographic differences in loneliness scores at school. Significant effects were observed for gender (p = .011), while other demographic variables showed no significant associations.

### Association between wealth index and loneliness

Narrative results: The relationship between loneliness at school and wealth index was not significant (p > 0.05). In contrast, loneliness at home was significantly associated with wealth index (df = 1751, mean score difference = 152.406, F = 2.853, p = 0.023). Post hoc analysis revealed a significant difference between the poorest (quintile 1) and middle-class households (quintile 3), with the poorest reporting higher loneliness (mean difference = 2.555, SD = 0.904, p = 0.048, 95% CI [0.015, 5.096]).

### Association between socioeconomic factors and loneliness

Table 5 shows that, access to electricity, piped water and gas was consistently associated with lower loneliness scores, particularly at home. Many household assets and structural indicators showed no significant relationship; certain utilities and resources stood out as important. Entertainment and communication resources showed mixed results. Television ownership was significantly related to reduced loneliness at home, pointing to its role in providing shared family experiences or a sense of connectedness, whereas radio, cell phones, bicycles and motor vehicles did not demonstrate significant effects.

### GEE analysis confirming predictors of loneliness across home and school settings

From Supplementary Table 3, the significant main effects were observed for gender, age, education and marital status: males

**Table 4.** Demographic distribution across loneliness score at home

| Variable | Group | N | Mean | df | t | Std. deviation | Std. error | p value | 95% CI lower | 95% CI upper |
|---|---|---|---|---|---|---|---|---|---|---|
| Gender | Female | 1,003 | 23.2134 | 1838 | 3.613 | 12.43612 | 0.39268 | **p < 0.001** | 0.9511 | 3.21039 |
| | Male | 837 | 21.1326 | | | 12.14162 | 0.41968 | | | |
| Age | 14–18 | 351 | 21.63 | 1817 | −1.122 | 11.78 | 0.629 | 0.262 | −2.258 | 0.614 |
| | 19–25 | 1,468 | 22.45 | | | 12.45 | 0.325 | | | |
| Religion | Protestant | 787 | 22.7408 | 1768 | 1.213 | 12.1705 | 0.43383 | 0.304 | 21.8892 | 23.5924 |
| | Catholic | 767 | 21.9674 | | | 12.42945 | 0.4488 | | 21.0864 | 22.8484 |
| | Muslim | 97 | 20.5567 | | | 12.98554 | 1.31848 | | 17.9395 | 23.1739 |
| | Others | 118 | 22.7373 | | | 12.47248 | 1.14819 | | 20.4634 | 25.0112 |
| Employment | No | 1,447 | 22.0435 | 1855 | 1.563 | 12.20826 | 0.32094 | 0.196 | 21.414 | 22.6731 |
| | No, but volunteering | 212 | 22.3066 | | | 12.95363 | 0.88966 | | 20.5528 | 24.0604 |
| | Yes, working full time | 67 | 23.7313 | | | 10.98702 | 1.34228 | | 21.0514 | 26.4113 |
| | Yes, Working part-time | 130 | 24.2231 | | | 13.26256 | 1.1632 | | 21.9216 | 26.5245 |
| Education | Primary | 154 | 25.2338 | 1858 | 7.528 | 10.70019 | 0.86225 | **p < 0.001** | 23.5303 | 26.9372 |
| | Secondary | 968 | 22.9525 | | | 12.16543 | 0.39101 | | 22.1851 | 23.7198 |
| | Tertiary | 609 | 20.7011 | | | 12.73862 | 0.51619 | | 19.6874 | 21.7149 |
| | University | 128 | 21.3672 | | | 12.69315 | 1.12193 | | 19.1471 | 23.5873 |
| Marital status | Never married | 1,131 | 21.6012 | 1852 | 2.474 | 12.46791 | 0.37073 | **0.022** | 20.8738 | 22.3286 |
| | In a relationship, but not living together | 408 | 22.8578 | | | 11.73957 | 0.5812 | | 21.7153 | 24.0004 |
| | Married/cohabiting | 204 | 24.0294 | | | 12.82313 | 0.8978 | | 22.2592 | 25.7996 |
| | Separated but not divorced | 36 | 27.0833 | | | 11.50497 | 1.91749 | | 23.1906 | 30.9761 |
| | Divorced | 11 | 25.1818 | | | 7.69179 | 2.31916 | | 20.0144 | 30.3492 |
| | Widowed | 7 | 19.5714 | | | 12.59441 | 4.76024 | | 7.9235 | 31.2193 |
| | Others | 56 | 22.1786 | | | 13.05806 | 1.74496 | | 18.6816 | 25.6755 |

*Note:* Independent-samples t-tests and one-way ANOVAs examined demographic differences in loneliness scores at home. Significant effects were observed for gender (p = .011), education (p < 0.001) and marital status (p = 0.022) while other demographic variables showed no significant associations. Post hoc analysis showed that participants with primary (vs. tertiary, p < .001) and secondary (vs. tertiary, p = .002) education reported significantly higher loneliness scores. No significant post hoc differences were observed for marital status.

reported higher loneliness than females (B = 2.009, SE = 0.60, p = 0.001); participants aged 14–18 reported lower loneliness than those aged 19–25 (B = −1.73, SE = 0.80, p = 0.031); participants with primary education had higher loneliness compared to those with university education (B = 3.99, SE = 1.55, p = 0.010) and marital status overall was significant (p = 0.035); however, individual comparisons revealed that no single category reached statistical significance, although participants who were married/cohabiting or separated tended to have higher loneliness scores.

A significant interaction between loneliness setting (home and school) and employment status was observed (Wald $\chi^2$ = 9.3, p = 0.023). Specifically, participants who were volunteering reported higher loneliness at home compared to school (B = 3.488, SE = 1.311, p = 0.008), and those in full-time employment also showed higher loneliness at home (B = 4.906, SE = 1.625, p = 0.003) (see Supplementary Table 3).

## Discussion

The experience of loneliness among youths is shaped by multiple social, economic and environmental factors, which may vary across contexts such as school and home. Examining these patterns and their association with economic and sociodemographic characteristics provides important insights into the distribution and determinants of loneliness, and helps to contextualize the findings within the broader social and developmental landscape of Kenyan youths living in the Nairobi metropolitan.

In our study, students reported higher levels of loneliness at school, as reflected in the greater frequency of responses such as "Often" and "Sometimes" across multiple items, compared to the home setting, where "Never" responses were more common. This aligns with global research identifying school as a context where social stressors such as peer exclusion, identity formation and academic pressure contribute to heightened loneliness among adolescents and young adults (Barreto et al., 2021; Walsham et al., 2023). From existing literature in Kenya and Africa, there is a general agreement that there is relatively lower loneliness at home than in schools (Branje and Morris, 2021; Adedeji et al., 2023). Our findings suggest the possibility that shifting family dynamics, urbanization and reduced communal support structures have made the school less protective for many young people, for example, the high prevalence of bullying in Kenyan schools (Ndetei et al., 2024). This raises the question of whether school environments may have become less socially protective potentially reflecting perceived

**Table 5.** Association between socioeconomic factors and loneliness

| | Loneliness at school | | Loneliness at home | |
|---|---|---|---|---|
| Variable | t(df), p | Mean difference (SE) [95% CI] | t(df), p | Mean difference (SE) [95% CI] |
| 1. Electricity | **t(1157) = 2.27, p = .023** | 2.21 (0.97) [0.30, 4.11] | **t(603.90) = 2.84, p = .005** | 1.94 (0.68) [0.60, 3.28] |
| 2. Radio | t(1157) = 1.27, p = .204 | 0.97 (0.76) [−0.52, 2.45] | t(1791) = 1.40, p = .161 | 0.84 (0.60) [−0.33, 2.01] |
| 3. Television | t(706.43) = 1.91, p = .056 | 1.46 (0.76) [−0.04, 2.96] | **t(1792) = 2.14, p = .032** | 1.34 (0.63) [0.11, 2.57] |
| 4. Refrigerator | t(248.19) = −0.49, p = .628 | −0.52 (1.08) [−2.65, 1.60] | t(1789) = 0.46, p = .649 | 0.40 (0.87) [−1.31, 2.10] |
| 5. Cell phone | t(1158) = −0.15, p = .877 | −0.13 (0.87) [−1.85, 1.58] | t(864.23) = 1.17, p = .244 | 0.74 (0.64) [−0.51, 2.00] |
| 6. Bicycle | t(1157) = −1.95, p = .051 | −1.81 (0.93) [−3.63, 0.01] | t(1790) = 1.27, p = .204 | 1.00 (0.78) [−0.54, 2.53] |
| 7. Motor cycle | t(1156) = −0.14, p = .890 | −0.15 (1.11) [−2.33, 2.03] | t(1789) = 1.48, p = .138 | 1.38 (0.93) [−0.44, 3.20] |
| 8. Motor vehicle | t(1154) = −1.27, p = .203 | −1.44 (1.13) [−3.65, 0.78] | t(1788) = −0.44, p = .657 | −0.45 (1.01) [−2.44, 1.54] |
| 9. Piped water | **t(1152) = 2.06, p = .039** | 2.02 (0.98) [0.10, 3.95] | **t(1787) = 3.12, p = .002** | 2.48 (0.79) [0.92, 4.04] |
| 10. Rain water | t(1151) = 0.47, p = .636 | 0.45 (0.95) [−1.41, 2.30] | t(1784) = 0.04, p = .967 | 0.03 (0.78) [−1.50, 1.57] |
| 11. Well water | t(1151) = 1.36, p = .173 | 1.22 (0.90) [−0.53, 2.98] | **t(1786) = 2.28, p = .023** | 1.73 (0.76) [0.24, 3.21] |
| 12. Surface water | t(1154) = −0.72, p = .474 | −0.77 (1.07) [−2.87, 1.33] | t(1787) = −0.47, p = .636 | −0.46 (0.97) [−2.36, 1.45] |
| 13. Other water sources | t(1155) = 0.69, p = .490 | 0.61 (0.88) [−1.11, 2.32] | t(1788) = 0.65, p = .514 | 0.50 (0.76) [−0.99, 1.99] |
| 14. Earth floor | t(1153) = −0.72, p = .471 | −0.74 (1.03) [−2.77, 1.30] | t(1786) = −0.24, p = .814 | −0.19 (0.81) [−1.77, 1.39] |
| 15. Cement floor | t(1156) = 0.85, p = .396 | 0.90 (1.05) [−1.16, 2.96] | t(1786) = −0.09, p = .930 | −0.07 (0.79) [−1.62, 1.48] |
| 16. Tile floor | t(1158) = 0.04, p = .966 | 0.04 (0.91) [−1.74, 1.82] | t(1788) = 0.75, p = .456 | 0.62 (0.82) [−0.99, 2.23] |
| 17. Wooden floor | t(1156) = 0.60, p = .546 | 0.63 (1.05) [−1.44, 2.70] | t(1787) = 0.84, p = .402 | 0.71 (0.85) [−0.96, 2.38] |
| 18. Other type of floor | t(1155) = 0.03, p = .976 | 0.03 (0.91) [−1.75, 1.81] | t(1785) = −0.35, p = .727 | −0.30 (0.86) [−1.99, 1.39] |
| 19. No toilet | t(1157) = −0.94, p = .348 | −0.96 (1.02) [−2.97, 1.06] | t(1787) = −0.28, p = .778 | −0.24 (0.85) [−1.91, 1.43] |
| 20. Pit latrine | t(1156) = 0.85, p = .394 | 0.85 (1.00) [−1.12, 2.82] | t(1786) = 1.06, p = .291 | 0.82 (0.78) [−0.71, 2.35] |
| 21. Flush toilet | t(1,031.14) = 0.18, p = .857 | 0.13 (0.75) [−1.33, 1.60] | t(1791) = −0.01, p = .992 | −0.01 (0.60) [−1.17, 1.16] |
| 22. Other type of toilet | t(1156) = −1.85, p = .065 | −3.26 (1.77) [−6.73, 0.20] | t(1790) = −1.67, p = .094 | −2.35 (1.40) [−5.09, 0.40] |
| 23. Firewood | t(1153) = −1.04, p = .297 | −0.89 (0.85) [−2.57, 0.78] | t(1787) = 0.79, p = .429 | 0.57 (0.72) [−0.84, 1.97] |
| 24. Charcoal | t(1152) = 1.75, p = .080 | 1.32 (0.76) [−0.16, 2.81] | t(1787) = 1.68, p = .093 | 1.02 (0.60) [−0.17, 2.20] |
| 25. Kerosene | t(635.77) = −0.10, p = .924 | −0.07 (0.79) [−1.62, 1.47] | t(1787) = −1.14, p = .256 | −0.72 (0.63) [−1.96, 0.52] |
| 26. Gas | t(1153) = 1.95, p = .052 | 1.46 (0.75) [−0.01, 2.92] | **t(1789) = 2.85, p = .004** | 1.68 (0.59) [0.52, 2.83] |
| 27. Electricity | t(1154) = −0.01, p = .994 | −0.01 (1.02) [−2.00, 1.99] | t(1790) = −0.18, p = .861 | −0.15 (0.84) [−1.80, 1.50] |
| 28. Other sources of fuel | t(1152) = −1.34, p = .181 | −7.48 (5.58) [−18.43, 3.47] | t(1787) = −1.74, p = .082 | −5.96 (3.43) [−12.69, 0.77] |

*Note:* Variables 1–8 are assets possessed, items 9–13 sources of water available, 14–18 type of floor 19–22 type of toilet and 23–28, source of fuel. Values are independent-samples t tests reported as t(df), p, mean difference (SE) and 95% CI. Positive values indicate lower loneliness with economic variable; negative values indicate higher loneliness. p < .05 was considered significant. Bold values are significant.

increases in bullying and psychosocial stressors rather than objectively measured changes in school safety. This may be particularly relevant in boarding schools, where separation from family support could intensify vulnerability to stress and peer victimization. Therefore, there is a need for focused research and interventions at the school level.

Consistent with our paired t-test results, loneliness scores were higher at school than at home (t = 5.30, p < 0.001). The mean difference corresponded to a small-to-moderate effect size (Cohen's d = 0.16), indicating that this difference is not only statistically significant but also practically meaningful. This suggests that students' experiences of loneliness at school may be noticeable in their daily social interactions and underscores the potential value of school-based psychosocial interventions. These findings align with previous research showing that social context influences youth loneliness and highlight the importance of targeting school environments when designing interventions to improve social connectedness (Birrell et al., 2025).

Gender differences in loneliness were evident across both school and home settings, with females consistently reporting higher loneliness scores than males. This aligns with broader global and African research showing that females often report greater emotional sensitivity to social exclusion and interpersonal disconnect (Smit, 2021). Education level did not significantly predict loneliness at school, but it did at home, where individuals with lower levels of education (particularly primary and secondary) experienced more loneliness. This corresponds with literature suggesting that education can be a buffer against loneliness by providing access to social capital and

coping strategies (Balki et al., 2023). At the same time, our finding that individuals with tertiary education also reported notable levels of loneliness reflects global evidence that higher education, while offering protective resources, can introduce new stressors that undermine social connectedness, intensifying social isolation by limiting peer engagement and increasing emotional withdrawal, thereby contributing to heightened loneliness among students (Walsham et al., 2023).

Marital status was not a significant factor at school but emerged as a key variable at home, with separated, divorced and married individuals reporting higher loneliness than those who had never married. This partially reflects global patterns that associate relational disruptions with increased loneliness (Kislev, 2022), but it also speaks to the Kenyan context, where shifts in family structure and weakening extended family ties may increase emotional isolation even within partnered households (Kathuri-Ogola and Kabaria-Muriithi, 2024). Stratified findings further suggest that as individuals transition into young adulthood (19–25 years), relational circumstances such as being married, divorced or separated and cohabiting play a key role in shaping loneliness at home experiences compared to adolescence (14–18 years), where peer networks dominate. Similar age-differentiated patterns have been observed in prior research (Brink, 2023). It is noteworthy that employment status did not significantly affect loneliness in either setting, contrasting with studies that have framed employment as a source of structure and social integration (Bryan, 2024). This finding should be interpreted in light of the study population, which comprised predominantly students and young people engaged in informal, work. We did not measure employment quality, stability and workplace social climate which may be more relevant determinants of loneliness than employment status alone in this context. This divergence may suggest that the protective effects of employment are context-dependent and influenced by job quality, social dynamics in the workplace and broader socioeconomic conditions (Dimitriu, 2022). Taken together, the results highlight how demographic factors interact differently across environments and reinforce the need to understand loneliness as a multidimensional experience shaped by cultural, structural and situational influences.

Although the overall effects of demographics were clearer at home than at school, the interaction analyses add nuance. Gender moderated the effects of marital status and employment on loneliness at home, suggesting that relational and occupational circumstances may affect women and men differently in domestic contexts. Similar gender-specific effects of marital status on loneliness have been reported in India (Pathak and Dewangan, 2025), aligning with our findings. However, other studies have found limited moderating role of gender in loneliness, highlighting that these interactions may vary across cultural and social contexts (Carney et al., 2020; Barreto et al., 2021). The absence of significant interactions at school underscores that educational environments might shape loneliness more uniformly across genders.

The GEE analysis, which accounted for repeated measures across school and home settings and adjusted for age, education, marital status and employment, largely confirmed these patterns while providing additional explanation. Age and education effects were consistent: younger participants reported lower loneliness than young adults, and participants with primary education had higher loneliness compared to those with university education. Marital status was significant overall, although individual categories were mostly nonsignificant; married/cohabiting and separated participants tended to report higher loneliness, aligning with the home-based patterns seen in the t-tests. The gender effect shifted in the multivariable model: males reported slightly higher loneliness than females, suggesting that the raw gender differences observed descriptively were partly influenced by other sociodemographic factors. A significant interaction between setting and employment indicated that participants who were volunteering or in full-time employment experienced higher loneliness at home than at school, while part-time employment showed a nonsignificant trend in the same direction. These findings highlight how loneliness is shaped by both individual characteristics and the environment, reinforcing and refining the initial descriptive patterns (Gazendam et al., 2020; Augustsson, 2025).

Our findings that middle-class households had lower levels of loneliness than poorest households align with findings from South Africa, which indicate that perceived lower household economic status is associated with greater loneliness, whereas individuals who view their economic status as average tend to experience fewer episodes of loneliness. This underscores the influence of socioeconomic conditions on the experience of loneliness (Posel, 2025).

We also found a significant relationship between socioeconomic factors and loneliness, with basic utilities such as electricity, piped water and gas consistently associated with lower loneliness scores, particularly in the home environment, underscoring the importance of essential infrastructure in fostering social connectedness (Reed and Bohr, 2021). Conversely, a great number of household possessions and structural measures, such as flooring type and sanitation facilities, were not significantly correlated, indicating that not all socioeconomic status markers are direct contributors to loneliness (Wee et al., 2019). Resources of entertainment and communication produced contradictory findings: television ownership, usually watched by the family, was associated with a decrease in loneliness at home, which may occur through the sharing of family experiences or a feeling of connectedness (Fingerman et al., 2022). However, the increased availability of smartphones and access to social media has the potential to increase loneliness among family members and less loneliness across social contacts. Only further studies will unravel this complexity. From our study, Other resources like radios, cell phones (at this point in time), bicycles and motor vehicles did not show significant changes, suggesting that not all access and ownership leads to social or emotional well-being as has also been reported elsewhere (Nowland et al., 2018).

## Limitations and recommendations

This study was limited by its cross-sectional design, which restricts causal inference, and by the use of self-reported questionnaire that may have introduced social desirability bias. Focusing on youth within the Nairobi Metropolitan Area may limit generalizability to rural or other urban settings. Despite these limitations, the findings provide valuable insight into contextual, economic and sociodemographic influences on loneliness.

Additionally, the purposive inclusion of youths from selected urban and peri-urban neighborhoods and reliance on community mobilization may have introduced selection bias, potentially limiting the representativeness of the sample.

Based on the results, there is a need for schools to strengthen inclusivity, peer bonding and anti-bullying measures to reduce social exclusion, while families should be supported through initiatives that promote communication, empathy and emotional connectedness. Policy makers should treat loneliness as a youth mental health and social development concern, integrating it into education, health and community programs. Further research, particularly longitudinal and qualitative studies, is recommended

to explore causal relationships and capture the lived experiences of loneliness among diverse Kenyan populations. Also, given high social media use among Kenyan youth, this dimension deserves specific exploration. Future research should focus on examining how social media use, cyber interactions and time spent online influence loneliness.

## Conclusion

Our study demonstrates that loneliness among young people is a multidimensional experience that varies across social environments, economic and sociodemographic factors. School emerged as a key setting where loneliness was more pronounced, underscoring the role of peer relations, identity formation and academic stress in shaping social connectedness. In contrast, the relatively lower loneliness reported at home highlights the potential protective role of the family, although this is influenced by factors such as education and marital status. These findings align with global and African evidence, but they also highlight the specific socioeconomic dynamics of Kenyan urban communities. Economic conditions also shaped loneliness experiences. Youth from middle-class households and those with access to basic utilities such as electricity, piped water and gas reported lower loneliness levels, emphasizing the role of essential infrastructure in supporting social connectedness. However, not all indicators of wealth, such as flooring type, sanitation facilities or the number of possessions, were linked to loneliness, suggesting that material abundance alone does not ensure social connectedness. Addressing loneliness requires a holistic approach that combines school-based inclusivity programs, family-centered support and broader policy initiatives to strengthen social bonds and emotional well-being. Future research could also benefit from longitudinal designs, which would allow for the examination of changes in loneliness over time and improve understanding of how contextual factors across different settings influence these experiences.

**Open peer review.** To view the open peer review materials for this article, please visit http://doi.org/10.1017/gmh.2026.10172.

**Supplementary material.** The supplementary material for this article can be found at http://doi.org/10.1017/gmh.2026.10172.

**Data availability statement.** Requests for the data may be sent to the corresponding author.

**Author contribution.** DMN – conceptualization and drafting of the paper; VM – oversight of data collection; CM – oversight on ethics; EJ – statistical analysis and literature review; PN – draft review; SW – statistical analysis; DK – statistical analysis and literature review; KO – literature review and methodology; VO – field work during data collection and literature review; SM – field work during data collection and literature review; DT – methodology; DO –literature review; YK – methodology and conclusion; AS – critique of the manuscript; DM – conceptualization and critique of the manuscript. All authors read and approved the final manuscript.

**Financial support.** This study was funded by the National Institutes of Health (NIH), Grant/Award number: 5R01MH127571-02.

**Competing interests.** The authors declare no conflict of interest.

**Ethics approval and consent.** The authors assert that all procedures contributing to this work comply with the ethical standards of the relevant national and institutional committees on human experimentation and with the Helsinki Declaration of 1975, as revised in 2008. All procedures involving human subjects/patients were approved by the Nairobi Hospital Ethics Research Committee (approval No. TNH-ERC/DMSR/ERP/022/22). The

study obtained licensing from the National Commission for Science, Technology and Innovation (NACOSTI) license number NACOSTI/P/22/18097. Institutional approval from the colleges was done as data collection involved students within these institutions. This ensured compliance with institutional policies on research involving students, facilitated access to participants and reinforced ethical standards, including informed consent procedures and data protection measures. Additionally, for the county-level community sample, administrative permissions were sought from the relevant county-level offices in Kiambu and Nairobi counties to allow engagement with community participants. These approvals were crucial for ensuring adherence to ethical and administrative requirements across both institutional and community settings. Informed written consent/assent was obtained from participants before data collection commenced. For participants younger than the age of 18, informed written consent to participate was obtained from their parents or legal guardians.

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
