## [Reviewer Report]

Thank you for offering this opportunity to review this paper.

This study aimed to examine prevalence and correlates of loneliness at schools and homes among Kenian youth. The authors demonstrated that loneliness was higher in schools compared to homes. While gender was associated with loneliness in both settings, education level, marital status was associated with loneliness at home only. The findings are important evidence for intervention in African context.

The current Introduction covers previous findings from a wide range of age groups (eg older age or adults) and from various countries, thus it is difficult to understand what the gap is for African or Kenyan children. Please revise the Introduction so that the evidence gap around adolescent loneliness, including the rational for comparing school and home loneliness is clear.

The authors found that loneliness at school was significantly higher than loneliness at home, which could be stratified by gender to see if there are gendered patterns.

In the discussion, you could consider discussing whether the observed difference in loneliness scores at home and schools are meaningful difference.

Table 1 may be more interpretable by showing as a figure.

---

## [Reviewer Report]

Grammatical errors

For the location name, Choose one style and keep it consistent (usually: “Nairobi Metropolitan Area” on first use, then “Nairobi metropolitan area”).

Abstract

“Loneliness is a complex phenomenon, experienced differently depending on context and environment This study…” is missing punctuation between sentences.

Abstract says total sample is 1,972 but later reports paired t-test df that imply a smaller paired sample (e.g., t(1165)). Clarify paired-sample N (e.g., “Among participants with both school and home scores (n=1166)…”).

Abstract gives significance but not effect size. Add the effect sizr

INTRODUCTION

“Loneliness affect people…” should be affects.

You cite Ndetei et al. (2025) for “low academic performance” but the Ndetei et al. (2025) paper in your reference list is about resilience and mental health correlates (depression, hopelessness, loneliness, PTSD, suicidality). Either (a) cite a study that actually examined academic performance, or (b) rewrite the claim to match what Ndetei et al. (2025) covers.

You discuss “schools” vs “home” broadly, but in methods participants include college + community, including out-of-school youth.

OBJECTIVES

General objective says Nairobi Metropolitan Area, but specific objectives say “Nairobi metropolitan” (inconsistent capitalization).

METHODS

The authors say “population-based” and “cross-sectional” but also say adolescents from disadvantaged neighbourhoods were “purposively included”. Clarify the sampling strategy (e.g., purposive site selection + convenience sampling; or cluster sampling; etc.). “Population-based” is a strong claim, only use it if you truly sampled to represent the population.

PARTICIPANTS

Whe n you say thaat due to influence of substances or illiteracy” you mix cognitive capacity and literacy, this may sound stigmatizing. Consider rephrasing: “unable to complete the questionnaire due to intoxication at time of interview or inability to read the questionnaire language.

You say participants were recruited from colleges and general community, including in-school and out-of-school youth. Add recruitment details: where/how approached, consent flow, whether any incentives, and response rate (especially important for bias).

Precisely define how you measured “loneliness at school” vs “loneliness at home” (same scale repeated twice? different instructions?).

ETHICS

Ethics section is strong, but it would help to specify data protection steps (anonymization, device security, storage).

“Institutional approval from the colleges was done…” awkward phrasing.

RESULTS AND TABLES

Overall, this is directionally appropriate for the study aim (compare loneliness across contexts and identify associated factors). The big gap is robustness/validity checks, the extra analyses like Assumption checks + diagnostic evidence help you prove your choices are sound.

Predictors and interactions, you can strengthen the paper by modeling both settings together:Linear mixed-effects model (LMM) (or GEE) with:-Outcome: loneliness score,-Within-person factor: setting (school vs home), -Predictors: gender, age, wealth index, etc., -Key terms: setting × predictor (e.g., setting × gender) to test if the factor behaves differently at school vs home.

This is better than running many separate tests because it Uses all data in one coherent model and handles within-person correlation properly

Table 1: You present mean loneliness at school/home by groups, but do not clearly state whether these are weighted, adjusted, or raw means.

Paired t-test uses n≈1166 (df=1165), but overall N is larger (1972). Add a brief missing-data note: how many had both scores, and why others did not.

Table 2 - ANOVA

For ANOVA, you should report df for between/within and clarify what “Mean score difference” represents (it looks too large and may be a sum of squares or mean square confusion).

Where ANOVA is significant (e.g., age, marital status), there’s no mention of post-hoc comparisons. Add post-hoc method (Tukey, Bonferroni, etc.) or state you did not conduct post-hoc and why.)

If participants come from multiple colleges/schools/areas, observations may not be independent.

Do one of either Mixed models with random intercepts for school/site, or regression with cluster-robust standard errors. This will help prevent overstating significance.

Table 3: Interaction effects : Interaction table lists coefficients but not the reference categories in the table itself. Add a footnote: reference group for xxxxxx

Table 4: GLM Mdel: Factors affecting loneliness at school” shows B, SE, p, CI, but doesn’t specify if it’s linear regression, and what covariates are included simultaneously. You need to state: “multivariable linear regression including age, gender, xxxxx

If loneliness scale range is, say, 20–80, then B≈2–4 is meaningful; but readers need scale description and range in Methods.

DISCUSSION

“This begs the question…” is often misused; you mean “This raises the question…”.

You cite Walsham et al. (2023) for academic stress/loneliness. Consider whether this source is appropriate/credible . If journal credibility is unclear, replace with a more established education/psychology source on academic stress and loneliness.

“Employment status did not significantly affect loneliness… contrasting with studies…” , good point, but you should connect to your sample composition (many are students; job quality not measured).

Discussion suggests schools became “less secure” (bullying, stressors). Considering phrasing as hypothesis/interpretation: “may reflect…”, not a definite conclusion unless you measured school safety.

LIMITATIONS AND CONCLUSION

Limitations mention self-report and cross-sectional nature (good), but should explicitly mention potential selection bias from purposive inclusion/recruitment method.

You recommend qualitative follow-up; that’s strong, also suggest longitudinal follow-up if feasible since context differs across settings.

REFERENCES

Bryan (2024) – citation is incomplete: no journal, no university, no report series, no DOI/URL.

DIMITRIU (2022) – chapter citation is incomplete (missing editors, publisher, and full book details).

Pathak & Dewangan (2025) – book chapter citation missing editors/ISBN/DOI.

---

## [Editor Report]

Dear Author,

Your manuscript: “A Comparative Study on the Intensity of Loneliness among Kenyan Youth in School and Home Environments”, has now been reviewed,

---

## [Editor Report]

Dear Authors,

Your revised manuscript entitled “A Comparative Study on the Intensity of Loneliness among Kenyan Youth in School and Home Environments”, has been reviewed,